

# Periodontal condition in growing subjects with Marfan Syndrome: a case-control study

Nicolò Venza[1], Carlotta Danesi[1], Diego Contò[1], Francesco Fabi[2], Gianluca Mampieri[1], Federica Sangiuolo[2] and Giuseppina Laganà[1]

[1] Department of Clinical Sciences and Translational Medicine, University of Roma "Tor Vergata", Roma, Italia
[2] Laboratory of Medical Genetics, Department of Biomedicine and Prevention, University of Rome "Tor Vergata", Rome, Italy

## ABSTRACT

**Background**. Marfan's syndrome (MFS) is a systemic disorder of connective tissue caused by mutations in the extracellular matrix protein fibrillin-1. Orofacial characteristics may be useful in identification of the syndrome. Severe periodontitis is sometimes observed in MFS patients, but no in-depth information has been reported in Italian groups of growing subjects with MFS. The aim of this study was to analyze the periodontal condition on a group of growing subjects affected by MFS, in comparison with a typically developed control group.

**Methods**. A group of 16 subjects with diagnosed MFS were recruited from the Centre for Rare Diseases for Marfan Syndrome and Related Disorders of Tor Vergata University Hospital. The Marfan Group (MG) was compared with a Control Group (CG) composed by 20 nonsyndromic subjects. The periodontal clinical parameters like Marginal Gingival Thickness (GT), Plaque Index (PI), Bleeding On Probing (BOP) and Modified Periodontal Screening and Recording (PSR) were assessed.

**Results**. The mean value of PI in MG was 59%, instead in CG it reached 21%. Analysis showed a significant difference between MG and CG also for the BOP. In MG the mean value of BOP attained 36% and in CG it reached 16%. A statistical significant difference of distribution of PSR index between the two groups was found for all sextant examined.

**Discussion**. Patients with Marfan syndrome reveal a higher presence of plaque and consequently a generalized inflammation in the oral cavity when compared with a control group.

## INTRODUCTION

Marfan's syndrome (MFS) is a systemic disorder of connective tissue caused by mutations in the extracellular matrix protein fibrillin-1; its prevalence has been estimated as 1 in 5–10,000 individuals (*Pepe et al., 2016*). MFS was firstly described in a 5-year-old girl by the French pediatrician Antoine Marfan in 1896 (*Marfan, 1896*). Later, the peculiar features detected by Marfan were associated with ectopia lentis (i.e., dislocation of the ocular lens) and mitral valve disease. In 1955, McKusick (*McKusick, 1955*) described aortic dissection and regurgitation in patients with Marfan syndrome and classified it as a heritable disorder of

Corresponding author
Giuseppina Laganà,
giuseppina.lagana@uniroma2.it,
lnggpp01@uniroma2.it

connective tissue, affecting cardiovascular, ocular, and musculoskeletal systems. Its genetic cause was detected in 1991, when *Dietz et al. (1991)* reported that a specific mutation results in classic Marfan syndrome. Most cases of MFS take origins from mutations in the gene for fibrillin-1 (FBN1), located on chromosome 15q-21.1 (*Loeys et al., 2010*). According to the literature, FBN1 is the only major gene involved in Marfan Syndrome (*Cook et al., 2015*).

Clinical manifestations typically involve the cardiovascular, skeletal, and ocular systems. The diagnosis is based on a combination of major and minor clinical features, described by Berlin classification system in 1986 (*Beighton et al., 1988*). A revised version of Ghent criteria for MFS was released in 2010 (*Loeys et al., 2010*). The major life-threatening manifestation of MFS is aortic aneurysm with high risk of dissection. Individuals with MFS often have multiple abnormal cardiac valves with prolapse and regurgitation. Consequently, these individuals are at greater risk to contract bacterial infective endocarditis than those who do not suffer from any structural heart disease (*Suzuki et al., 2015a*).

Orofacial characteristics may be useful in the identification of the syndrome, consisting in: long and narrow face, maxillary/mandibular retrognatia, temporomandibular joint alterations, high arched palate, dental crowding, posterior crossbite and sleep disorders (*Bostanci, Korkut & Unlu, 2017*; *Paoloni et al., 2018*; *Laganà et al., 2018*). It is known that Marfan syndrome is an autosomal dominant disorder, affecting the elastic system fibers. The responsible gene for Marfan syndrome has been identified as FBN1, which encodes the major microfibrillar protein, fibrillin-1 that play a crucial role in the composition of elastic system fibers (*Dietz et al., 1991*; *Maslen et al., 1991*). MFS has been shown to increase the susceptibility to severe periodontal disease in association with periodontal ligament dysfunction due to microfibril insufficiency, suggesting that FBN-1 microfibril formation plays a central role in periodontal ligament formation. Notably, the elastic fibers of the periodontal ligament, known as oxytalan fibers, primarily consist of FBN-1 microfibrils and do not contain significant amounts of elastin. Therefore, the periodontal ligament is likely more susceptible than other connective tissues to breakdown in the MFS mouse model. Thus, periodontal disease is a useful model to assess the effect of MFS on inflammatory tissue destruction (*Handa et al., 2018*). Thus, MFS patients may be susceptible to periodonto-pathogenic bacteria that invade the periodontal area, causing connective tissue disorders too.

The periodontal ligament (PDL) is a specialized connective tissue composed of collagen and elastic system fibers and situated between the cementum covering the root of teeth and the alveolar bone socket (*Beertsen, McCulloch & Sodek, 1997*).

Furthermore, severe periodontitis is sometimes observed in MFS patients, but no in-depth information has been reported in growing subjects with MFS.

Previous studies revealed a strong relationship between periodontal diseases and pathophysiology of cardiovascular complications in MFS patients (*Suzuki et al., 2014*; *Suzuki et al., 2015a*; *Suzuki et al., 2015b*).

To our knowledge, few studies were conducted on animals (*Handa et al., 2018*) or adult subjects (*Staufenbiel et al., 2013*) but no study has been conducted on growing subjects with Marfan Syndrome analyzing the periodontal index. Considering the high risk of
**Table 1 Genetic and clinical characteristics of Marfan Group.**

| S | Age | Sex | MT | PD | AD Z-score | PC | R | LF |
|---|-----|-----|-----|-----|------------|-----|-----|-----|
| D.P. | 7 | M | Splice | cb EGF-like#36 | 2.26 | X | X | X |
| C.M. | 13 | M | FS | cb EGF-like#13 | 3.29 | X | | |
| C.C. | 7 | M | STOP | cb EGF-like #04 | 3.3 | X | X | X |
| C.G. | 14 | M | Missense | cb EGF-like #42 | 3.29 | X | X | X |
| D.M. | 11 | F | Splice | cb EGF-like#36 | 2.49 | | | X |
| T.M. | 5 | M | Missense | cb EGF-like #11 | 13.1 | X | | |
| R.A. | 9 | F | Splice | cb EGF-like#36 | 5.40 | X | X | X |
| D.A. | 9 | M | Splice | cb EGF-like#36 | 2.93 | X | | X |
| B.M | 8 | F | Splice | cb EGF-like#36 | 2.30 | | X | X |
| B.S. | 11 | M | Splice | cb EGF-like#36 | 2.90 | X | X | |
| D.A. | 7 | F | Splice | cb EGF-like#36 | 4.41 | X | X | X |
| G.A. | 12 | F | FS | cb EGF-like#33 | 3.57 | | | X |
| M.L. | 8 | M | Missense | cb EGF-like #41 | 3.72 | X | X | X |
| D.I. | 10 | F | Splice | cb EGF-like#36 | 2.74 | X | | |
| D.A. | 9 | M | Missense Cys | cb EGF-like #14 | 3.57 | X | | X |
| G.C. | 10 | F | FS | cb EGF-like#33 | 5.00 | X | | |

**Notes.**

S, Subjects; MT, Mutation type; PD, Protein domain; AD, Aortic dilatation; PC, Palatal constriction; R, Retrognatia; LF, Long face.

Marfan subjects to contract bacterial infective endocarditis, further studies concerning the periodontal and gingival conditions are necessary.

The aim of this study was to analyze the periodontal condition in a group of growing subjects affected by MFS, in comparison with a control group.

## MATERIALS & METHODS

The study protocol was approved by the Ethical Committee of the University of Rome Tor Vergata (Protocol number: 4544/2017) and an informed consent was signed by the parents of the patients involved.

For the actual preliminary study a group of 16 Italian subjects with genetic assessment of MFS, were recruited from the Centre for Rare Diseases for Marfan Syndrome and Related Disorders of Tor Vergata University Hospital, from November 2017 to April 2018 and, evaluated in the Departments of Orthodontics of the same Institution. The inclusion criteria were: genetic assessment of Marfan Syndrome, Caucasian ancestry, no post pubertal vertebral maturation (CS5–CS6).

The Control Group (CG) was composed by 20 subjects (8 males and 12 females) and was collected from the Departments of Orthodontics of Tor Vergata University Hospital, using the following inclusion criteria: Caucasian ancestry, good occlusion, no previous orthodontic treatment, no deciduous stage of dentition, no post pubertal vertebral maturation (CS5–CS6). All the subjects in the control group were non- syndromic subjects

Exclusion criteria for both MG and CG were: presence of orthodontic treatment, cleft lip and/or palate, other genetic diseases. Table 1 describe the Marfan gruop characteristics.

The study teeth included both maxillary and mandibular arches. Locally, teeth with restorations within 1 mm of the gingival margin, crowded, ectopically positioned, or with history of surgical procedures, were excluded.

A single examiner performed all clinical measurements using a plane-faced dental mirror and Michigan O probe with Williams markings coded at 1, 2, 3, 5, 7, 8, 9, and 10 mm. Intra-examiner reliability was performed by repeating the clinical measurements on three randomly-selected patients one week later.

The periodontal clinical parameters were assessed in the following order:

1.  Marginal gingival thickness (GT) was assessed by visibility of the out-line of the probe through the marginal gingiva (visible, thin, not visible, thick) (*Kan, Rungcharassaeng & Lozada, 2003*);
2.  The plaque index (PI) was recorded based on the scale of *Silness & Loe (1964)*. A periodontal probe was used, recording the variables in six localizations of each tooth;
3.  Bleeding on probing (BOP) were recorded using a manual probe at six points (buccal-mesial, mid-buccal, buccal-distal, lingual-mesial, mid-lingual, lingual-distal) on a right upper molar, an upper incisor, a left upper molar, a right lower molar, a lower incisor and a left lower molar;
4.  Modified Periodontal Screening and Recording (PSR) (*Landry & Jean, 2002*) indices were produced through oral health assessments. To record probing depths, the mouth was divided into sextants. Gingival crevices around each tooth were examined, and researchers recorded the deepest probing depth found in each sextant.

All statistical analyses were performed with the aid of statistical software (Statistical Package for Social Sciences, version 16.0, SPSS Inc., Chicago, USA). Values of $P < 0.05$ were considered significant. Descriptive statistics were used to describe both sample groups (SG and CG) in terms of age, sex and gingival thickness. Statistical significance of differences between groups for qualitative variables were assessed using the Chi-squared test; $t$-test for unpaired data was applied for assessing the comparison of the quantitative variables between groups.

## RESULTS

A total of 9 males and 7 females (mean age of $9.4 \pm 2.3$ years) with confirmed MFS were evaluated in the Department of Orthodontics of Tor Vergata University. 8 males and 12 females with a mean age of $10.0 \pm 2.6$ years composed the Control Group. About the Marginal gingival thickness, in the MG, 5 subjects showed thick marginal gingiva and 11 subjects showed thin marginal gingival thickness. Similar results were found in the CG, in which 6 subjects showed thick marginal gingiva and 15 subjects showed thin marginal gingival thickness.

The mean value of PI registered in the MG was 59% instead in the CG it reached 21%. $T$-test for unpaired data showed a statistical significant difference ($P < 0.05$) between the two groups.

Analysis showed a significant difference ($P < 0.05$) between MG and CG also for the BOP evaluated for all the subjects in both groups. In MG the mean value of BOP attained

**Table 2  Distribution of periodontal indexes compared with the control group.**

|  | Marfan group (n = 16) | Control group (n = 20) | t-student | Degree of freedom | P value | CI 95% |
|---|---|---|---|---|---|---|
| Plaque index | 59% | 21% | 8.23 | 24.64 | 0.000[*] | 0.28–0.47 |
| Bleeding on probing | 36% | 16% | 7.95 | 27.39 | 0.000[*] | 0.19–0.33 |
| Medial PSR | 1.69 | 0.39 | 5.36 | 20.86 | 0.000[*] | 0.85–1.92 |

**Notes.**
[*]Sig < 0.05, t-test for unpaired data.
PSR, Periodontal screening and recording.

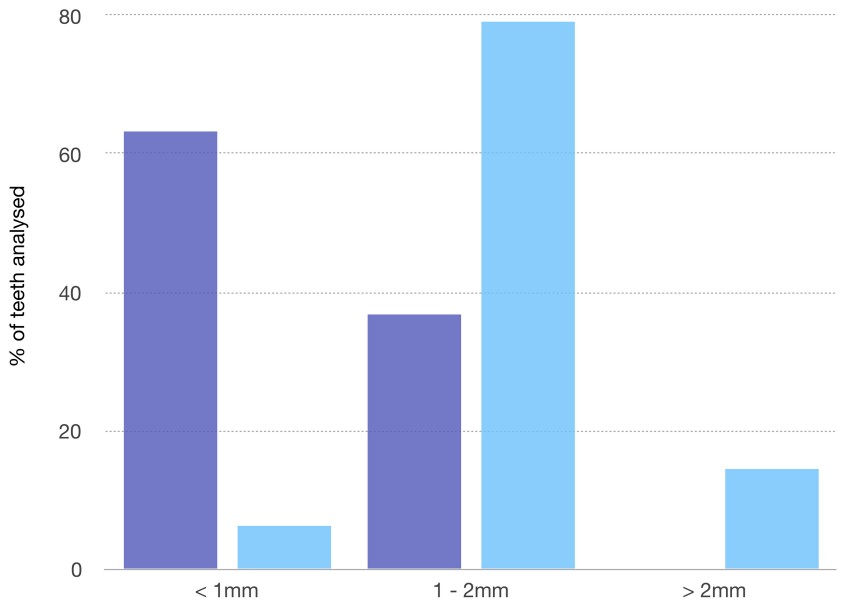

**Figure 1  PSR index recorded in the two groups.** The figure describes the distribution of PSR in the two groups.

36% and in CG it reached 16%. Table 2 describes the statistical analysis of PI, BOP and medial PSR.

Distribution of PSR index was reported in Figs. 1 and 2. A statistical significant difference between the two groups was found for all sextant examined.

## DISCUSSION

The current study examined the relationships among some periodontal disease indicators in a group of growing subjects affected by MFS, compared with a control group.

Several authors reported severe periodontitis in patients with other heritable disorders of connective tissue such as some types of Ehlers-Danlos syndrome (EDS), in particular in types III, IV, and VIII (*Karrer, Landthaler & Schmalz, 2000*).

It was well reported that periodontitis is frequently observed in MFS patients (*De Coster, Martens & De Paepe, 2002*; *Straub et al., 2002*; *Shiga et al., 2008*; *Suda et al., 2009*). This

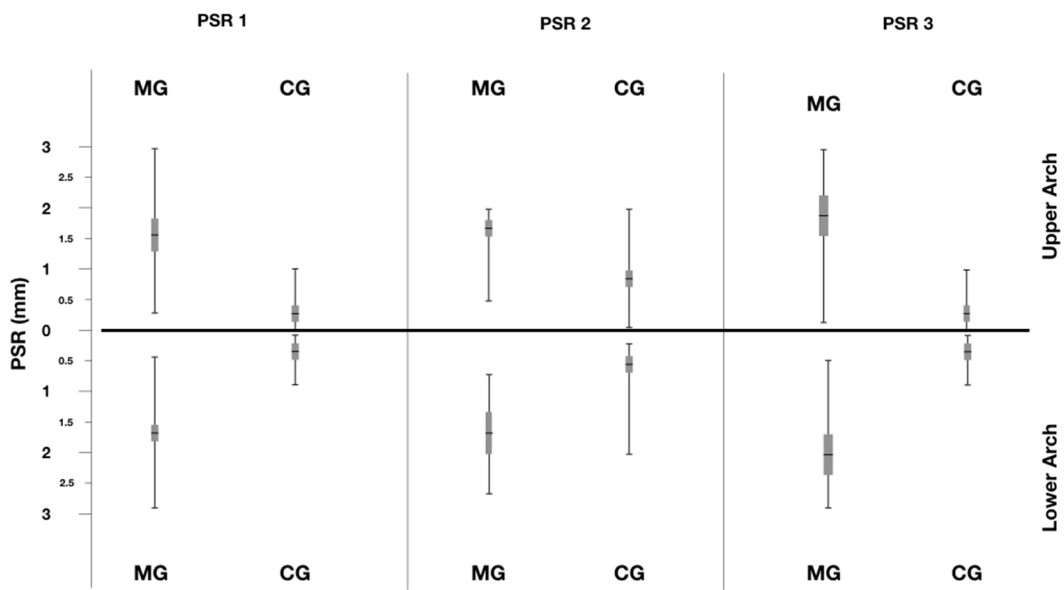

**Figure 2  Distribution of PSR index for the sextant examined in Marfan and Control Groups.**

genetic disorder causes structural and cellular defects that play a role in the inflammatory periodontal condition.

MFS patients may be susceptible to periodontopathic bacteria, that invade from the periodontal area with a connective tissue abnormality.

Moreover, periodontitis may influence the pathophysiology of cardiovascular complications in MFS patients (*Suzuki et al., 2016*). *Judge & Dietz (2008)* revealed that individuals with MFS were at high risk of developing infective endocarditis with dental disorders (*Judge & Dietz, 2008*). For this reason, further studies on periodontal conditions of Marfan subjects are recommended in order to establish optimized oral care programs and avoid possible infective systemic complications.

The present study analyzed for the first time the association of MFS and gingival inflammatory condition in young subjects.

*Straub et al. (2002)* in a case report of a 41-year-old patient who had generalized inflammation in the oral cavity revealed both bleeding and retentive factors such as bacterial plaque and tartar.

Our study, unlike the previous ones, considered a group of growing subjects, in order to evaluate possible early correlations between periodontitis and Marfan's syndrome. The results evidenced that already at an early age in MFS subjects a plaque index of 59% and a bleeding on probing attaining 36% is possible to be observed.

Periodontal disease among children and adolescents consists mainly of gingivitis (*Meyle & Gonzáles, 2001*). The prevalence of marked periodontal alteration is lower in young individuals. In the USA, the prevalence of severe periodontal attachment loss on multiple teeth among children and young adults is between 0.2% and 0.5%. Periodontal diseases

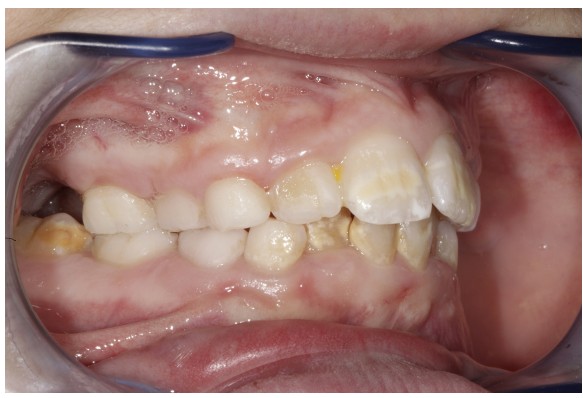

**Figure 3** **Periodontal condition in a growing subject with Marfan Syndrome.** Figure shows gingival conditions in a subject affected by Marfan Syndrome. Gingival tissue is typically hypertrophic in subjects with Marfan Syndrome. Photo by Dr. Giuseppina Laganà.

in young individuals can develop as a consequence of a local or a systemic factor (*Löe & Brown, 1991*).

In a study carried out by Suzuki et al. in 2013, that analyzed a group of 40 Marfan subjects and 15 control group's subjects, MS patients presented periodontitis more frequently than control group subjects at the same age. In addition, patients with MS presented significantly severe periodontitis, compared to control group subjects (*Suzuki et al., 2015b*).

The data from our study show that patients with Marfan syndrome have a gingival inflammatory condition statistically significant higher than patients in GC. Furthermore, the presence of pockets up to 4 mm has also been detected, especially on permanent teeth (Fig. 3).

Considering the increased risk of cardiovascular complications suffering from Marfan syndrome, early diagnosis is essential to the success of treatment and prevention of infection. To combat the severity and progression of periodontal disease among persons with these systemic diseases, a preventive strategy should be implemented that includes more frequent than 6 months professional oral hygiene, chlorhexidine mouth rinse, antibiotic therapy when indicated, fluoride gel, and/or professionally applied fluoride varnish (*Alrayyes & Hart, 2011*).

## CONCLUSIONS

To the best of our knowledge, this is the first study analyzing the periodontal index in a group of growing subjects affected by MFS. Based on our data, patients with Marfan syndrome reveal a higher presence of plaque and consequently a generalized inflammation in the oral cavity. Moreover, no signs of severe periodontal attachment loss on multiple teeth were on growing subjects with MFS.

## ACKNOWLEDGEMENTS

The authors wish to express their gratitude to all the staff of the Centre for Rare Diseases for Marfan Syndrome and related disorders.

### Funding

The authors received no funding for this work.

### Competing Interests

The authors declare there are no competing interests.

### Author Contributions

- Nicolò Venza conceived and designed the experiments, prepared figures and/or tables, approved the final draft.
- Carlotta Danesi performed the experiments, analyzed the data.
- Diego Contò performed the experiments.
- Francesco Fabi analyzed the data, prepared figures and/or tables.
- Gianluca Mampieri contributed reagents/materials/analysis tools, authored or reviewed drafts of the paper, approved the final draft.
- Federica Sangiuolo analyzed all the genetic data of the studied subjects.
- Giuseppina Laganà conceived and designed the experiments, contributed reagents/-materials/analysis tools, authored or reviewed drafts of the paper, approved the final draft.

### Human Ethics

The following information was supplied relating to ethical approvals (i.e., approving body and any reference numbers):

The study protocol was approved by the Ethical Committee of the University of Rome Tor Vergata (Protocol number: 4544/2017) and an informed consent was signed by the parents of the patients involved.

### Data Availability

The raw measurements are available in the Supplemental File. The raw data show all the measurements taken and the statistical analysis.

### Supplemental Information

Supplemental information for this article can be found online at http://dx.doi.org/10.7717/peerj.6606#supplemental-information.

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
