# Peer review of "Periodontal condition in growing subjects with Marfan Syndrome: a case-control study"

_PeerJ, doi:10.7717/peerj.6606_

## Round 0.1 · original submission · Major Revisions

Dear authors

Your ms has been evaluated and the reviewers suggested to modify several things in your ms and to supplement it with new information.

·

Basic reporting

In their work Venza et al report the association between Marfan syndrome (MFS) and increased periodontal conditions, namely plaque and inflammation, in paediatric patients.

The language is acceptable, although different grammar mistakes are present in the text (e.g. lines 125, 132: “founded” is not the past tense of “find” – “found” should be used; line 57: “do not” should be used, and not “don’t”; line 65: the whole sentence is not properly formulated).
The introduction properly inserts the study in the actual situation, reporting what is currently known concerning the bases of MFS, and its correlation with periodontal diseases. The authors should improve the description of the relevance of their work already at the end of the introduction section.
The three figures provided could be simply merged in one panel. Figure 3 is of very poor quality, and it is not suitable for publication in big scale: this should be either changed, or published in a much smaller size.
The discussion is poor. The authors fail to show the relevance of their study for any clinical application. For example, the prognostic value of their finding is not discussed at all, as well as the potential importance of earlier treatment/preventive approaches in MFS patients before the onset of more severe periodontal affections.

Experimental design

The authors should improve the description of the relevance of their work already at the end of the introduction section. At line 73, the specification of the nationality of the patients does not really increase the impact of the manuscript, and it is not relevant at this point. Similarly, the statement that “the causal relationships among MFS, aortic aneurysms, and periodontitis have not been elucidated yet” (line 76-77) at the very end of the introductory paragraph does not really fit, as this is not the central aim of the manuscript. The authors should focus more on the only aspect of novelty of their article, that is, the study of the correlation between MFS and periodontal conditions in young patients.

The experimental design is properly described. The study is extremely simple, as it consists of a reduced set of clinical observations in MFS and control age-matched patients. The results are properly described.

Validity of the findings

The data appear solid, even though all based on subjective clinical evaluations.
The conclusions are solid, while very poor is their discussion in the light of their potential implications for the current clinical practice.

Additional comments

no comment

Reviewer 2 ·

Basic reporting

English needs some revision being unclear in some sentences.
For example: lines 79, 146-147.

The article includes sufficient introduction which is unprecise/not correct in one point (lines 47-49). FBN1 gene is at the moment the only gene associated to Marfan syndrome

Litterature is mainly appropriately referenced except for one point where is inappropriately referenced (lines 250-251). (substitute Summers with Cook JR, Carta L, Galatioto J, Ramirez F. Cardiovascular manifestations in Marfan syndrome and related diseases; multiple genes causing similar phenotypes. Clin Genet. 2015;87(1):11–20 to show that FBN1 is the only major gene involved in Marfan syndrome).

An important Table reporting clinical features that allow the diagnosis of Marfan syndrome and the statistical P value for the plaque index (PI), bleeding on probing (BPO), medial periodontal screening and recording (PSR) for all the 16 patients is missing.

When the Authors say . “severe periodontitis is sometimes observed in MFS patients, but no in-depth 
information has been reported in Italian groups of MFS growing subjects” (lines 72-73) do they mean that “in depth information” have been reported for different geographic groups? If yes, please, specify which groups and add references/, otherwise erase the adjective “Italian” to avoid misunderstanding.

Experimental design

If, as the Authors say, severe parodontitis has been already reported in Marfan patients the Authors recruited 16 patients from the Centre for Rare Diseases for 
Marfan Syndrome and Related Disorders of Tor Vergata University Hospital, from November 
2017 to April 2018. with the aim of investigating early signs of periodontitis in growing patients (age: 7-12 years). The number of recruited patients in 6 months is pretty low for this purpose in fact, at the end of the screening, the informative patients are 5. Five patients out of 16 (about 30%) display early signs.

In the experimental design there is an important weak point: the Authors base the diagnosis of Marfan syndrome on the clinical manifestations (but they do not show the clinical features). It is very difficult to make the diagnosis at 7-12 years without confirmation by mutation analysis.

It is not clear what the Authors mean by "controls made of 20 non-syndromic subjects" (line 90) and the exclusion criteria they use for both MG and CG (lines 91.92) are not clear also.
Controls are not supposed to have a systemic features score >4. If the controls display systemic features of Marfan syndrome this should be reported.

It is not clear if these are the first data reporting early signs among world MF patients or among Italian Marfan patients, please clarify.

Validity of the findings

Results.

The main result which consists in finding an altered periodontal index and a higher presence of plaque and consequently a generalized inflammation in the oral cavity of 5 adolescent Marfan patients is made weak by the absence of a Table displaying the clinical features that allowed the diagnosis of Marfan syndrome, the absence of data on related FBN1 pathogenetic mutations, the absence of statistical data for each of the 5/16 patients with parodontitis, the very small number of patients.

If the Reviewer understands well 30% of patients and controls display thick marginal gingival. Then, they affirm that three clinical parameters show the presence of early parodontitis in 5 Marfan patients.The statistical P value for the plaque index (PI), bleeding on probing (BPO), medial periodontal screening and recording (PSR) should be reported for all the 16 patients.

Discussion.

The Authors say "Ganburged and coworkers revealed that capillaries were dilated or enlarged in PDL cells of mice 
underexpressing fibrillin-1 (Ganburged et al., 2010). Thus, MFS patients may be susceptible to 
periodontopathic bacteria, that invade from the periodontal area with a connective tissue 
abnormality." 

In Gambured et al 2010 capillaries resulted dilated in mice homozygous, not heterozygous, for FBN1 mutations. Marfan patients are heterozygous for FBN1 mutations. Therefore, the Authors cannot state the above correlation.

Conclusions (lines 182-183)
Authors write "to the best of our knowledge, this is the first study analyzing the periodontal index in a group of 
Italian growing subjects affected by MFS". 


It is a result based on a very small number of patients (5/16) displaying these early features. It is not clear to the Reviewer if these data are the first reported in Italian Marfans or in world Marfans. In the first case the Authors have to mention the articles reported for the other Marfan patients and compare the data.

In conclusion this article has some important weak points:
1. A table with the clinical features allowing the
clinical diagnosis of the 16 MF patients is
missing.
2. Also a table (or the same table) with the statistical
results regarding the three dental parameters is
missing.
2. FBN1 mutational analysis is not reported
3. Patients are too young to be definitly diagnosed
affected by Marfan syndrome on the bases of
clinical manifestations unless they are strongly
direct towords this diagnosis
4. The results are based on a total of 5 patients
which is a poor number

---

## Round 0.2 · Major Revisions

Dear authors

Your ms has been reviewed and one of the reviewers request an explanatory table for accepting your publication. If you cannot provide it I am afraid that your ms will not be accepted for publication.
Therefore, a Table reporting the major clinical data of the 16 patients which demonstrates the correct clinical diagnosis should be provided.

·

Basic reporting

-

Experimental design

-

Validity of the findings

-

Additional comments

The authors satisfactorily addressed the comments and issues raised during the first round of revision. I therefore consider the manuscript ready for publication.

Reviewer 2 ·

Basic reporting

Abstract. Page 1. Line 25: "non-syndromic subjects" do not represent a correct "control group".

Introduction. Page 2. Lines 49-51. It seems that the Authors mixed up the Berlin nosology with the Ghent nosology. It would be better to erase "Ghent nosology" and substitute the reference "Ha et al 2007" with the original one of Beighton and coworkers. (1988)

Experimental design

Are you saying that 16/16 patients were positive for FBN1 gene mutation? It is a very high score, 100%. Maybe you selected these patients with positive FBN1 mutation among a larger group of young patients.

Validity of the findings

Results. In the reviewer opinion results have to be based on a correctly selected group of patients. The reviewer is confused by the fact that the Authors do not report, as previously requested, a table with at least major clinical data on each patient confirming, together with genetic data, the clinical diagnosis.
This reviewer believes that a simple Table displaying the clinical manifestations of each patient (ocular, CVS, SF) that, together with the genetic data, brings to the diagnosis of Marfan syndrome is necessary.

---

## Round 0.3 · accepted · Accept

The ms is now accepted for publication